# Microbial-Based Biofungicides Mitigate the Damage Caused by *Fusarium oxysporum* f. sp. *cubense* Race 1 and Improve the Physiological Performance in Banana

**DOI:** 10.3390/jof10060419

**Published:** 2024-06-12

**Authors:** Luisa Fernanda Izquierdo-García, Sandra Lorena Carmona-Gutiérrez, Carlos Andrés Moreno-Velandia, Andrea del Pilar Villarreal-Navarrete, Diana Marcela Burbano-David, Ruth Yesenia Quiroga-Mateus, Magda Rocío Gómez-Marroquín, Gustavo Adolfo Rodríguez-Yzquierdo, Mónica Betancourt-Vásquez

**Affiliations:** Corporación Colombiana de Investigación Agropecuaria, AGROSAVIA, Centro de Investigación Tibaitatá, Km 14 vía Bogotá a Mosquera, Mosquera, Cundinamarca 250047, Colombia; scarmona@agrosavia.co (S.L.C.-G.); avillarreal@agrosavia.co (A.d.P.V.-N.); dmburbano@agrosavia.co (D.M.B.-D.); ryquiroga@agrosavia.co (R.Y.Q.-M.); mrgomez@agrosavia.co (M.R.G.-M.); grodriguezy@agrosavia.co (G.A.R.-Y.); mbetancourtv@agrosavia.co (M.B.-V.)

**Keywords:** biological control, microbial consortia, bioproducts, Panama disease

## Abstract

Fusarium wilt of banana (FWB) is the most limiting disease in this crop. The phytosanitary emergency caused by FWB since 2019 in Colombia has required the development of ecofriendly control methods. The aim of this study was to test the effectiveness of microbial-based biofungicides against FWB caused by *Fusarium oxysporum* f. sp. *cubense* race 1 (Foc R1) and correlate such effect with plant physiological parameters. Five *Trichoderma* (T1 to T4 and T9) and four *Bacillus* (T5 to T8)-based biofungicides were evaluated in pot experiments. In vitro, dual confrontation tests were also carried out to test whether the in vitro effects on Foc growth were consistent with the in vivo effects. While *Trichoderma*-based T3, T4, and T9, and *Bacillus*-based T8, significantly reduced the growth of Foc R1 in vitro, *Trichoderma*-based T1, T3, T4, and T9 temporarily reduced the Foc population in the soil. However, the incidence progress of FWB was significantly reduced by Bacterial-based T7 (74% efficacy) and *Trichoderma*-based T2 (50% efficacy). The molecular analysis showed that T7 prevented the inner tissue colonization by Foc R1 in 80% of inoculated plants. The T2, T4, T7, and T9 treatments mitigated the negative effects caused by Foc R1 on plant physiology and growth. Our data allowed us to identify three promising treatments to control FWB, reducing the progress of the disease, delaying the colonization of inner tissue, and mitigating physiological damages. Further studies should be addressed to determine the modes of action of the biocontrol agents against Foc and validate the utilization in the field.

## 1. Introduction

The banana is one of the most consumed and traded foods worldwide [1]. This fruit is part of the family basket in the countries where it is grown, due to its nutritional characteristics, thus playing an essential role in food security [2] (pp. 15–19), [3] (pp. 73–76), [4]. On the other hand, banana exports from developing countries substantially contribute to their economic income [2].

However, worldwide, banana-producing farmers have had to face the threat of fungal diseases to fruit production throughout history. The Fusarium wilt of banana (FWB), also known as Panama disease, caused by *Fusarium oxysporum* f. sp. *cubense* (Foc), is the most destructive and difficult to control [5,6]. Three races of Foc have been identified according to their ability to cause disease in specific banana cultivars. Race 1 (Foc R1) affects banana varieties such as Gros Michel (AAA), Manzano (AAB), and Pisang Awak (ABB). Race 2 (Foc R2) affects cooking bananas, especially those of the Bluggoe (ABB) subgroup. Lastly, race 4 (Foc R4) affects all those varieties susceptible to Foc R1 and Foc R2, especially cultivars of the Cavendish subgroup [7,8].

*F. oxysporum* f. sp. *cubense* Tropical Race 4 (Foc TR4) has spread inexorably through the banana-producing regions around the world, including South America, where symptomatic plants from the cv. Cavendish, resistant to Foc R1, were detected in 2019 [9]. The same year, Colombia’s national plant protection organization (Instituto Colombiano Agropecuario—ICA) declared a phytosanitary emergency, since Foc TR4 was isolated from Cavendish plantations located in the north of the country (i.e., La Guajira Department) showing typical symptoms of wilt disease [10]. This phytosanitary problem has been treated as a quarantine disease, and the official institutions, in cooperation with banana growers, are working together to contain and eradicate the pathogen in the affected area, implementing dispersal prevention measures and developing control methods [11].

The typical external symptoms of FWB include yellowing of the leaf blade, pseudostem breakage, and wilting of the oldest leaves [12]. These symptoms are caused by the obstruction of the vascular bundles exerted by the proliferation of Foc structures and the formation of gums and tyloses secreted by the plant in the vascular tissue. This plugging increases the resistance in the translocation of water and nutrients and decreases the water potential in the leaves. As a consequence, stomatal conductance decreases and, by reducing CO_2_ absorption, the photosynthetic activity and quantum efficiency of photosystem II (PSII) are limited, ultimately affecting biomass accumulation and plant growth [13,14].

The dispersal of Foc TR4 is particularly difficult to control since this fungus spreads easily through contaminated soil, water, and agricultural tools [15]. To deal with this pathogen, the research has been focusing on diagnosis and early detection, containment of the pathogen through the destruction of infected plants and declaration of quarantine areas, implementation of biosafety schemes [7,16], development of resistant varieties [17,18], the implementation and application of biological control [19,20,21,22,23], and crop management practices [24,25,26,27].

Considering that the indiscriminate use of chemical pesticides to control plant pathogens has caused a negative impact on the environment [28,29], biological control as an alternative tool for the management of Fusarium wilt is projected as a very effective and environmentally friendly strategy [30]. Biological control with antagonistic fungi and bacteria mainly from the genera *Trichoderma* and *Bacillus*, respectively, has been focused on reducing the negative effects caused by soil-borne phytopathogens on crop yields [20,31,32] through the stimulation of plant defense mechanisms or exerting direct mechanisms of action on the target phytopathogens. In all of this, complex interactions between plant-pathogen antagonists and the resident microbiota take place.

Several well-known species of fungi from the genus *Trichoderma* have been recognized for their high biocontrol potential against diseases and insect pests, as was shown in the recent review of Sharma et al. [33]. Accordingly, different *Trichoderma* species have the ability to multiply rapidly and advantageously inhibit the growth of fungi through the production of different enzymes (viz., glucanases, chitobiosidases, and chitinases) used mainly during the parasitizing action on pathogenic fungi; low-molecular-weight volatile or nonvolatile antibiotics or secondary metabolites (viz., viridin, gliotoxin, and peptaibols) that restrict the growth of pathogens; and the induction of local or systemic defense mechanisms in host plants. Some of the most known species of *Trichoderma* with antagonistic properties are *T. harzianum*, *T. asperellum*, *T. koningii*, *T. koningiopsis*, *T. longibrachiatum*, *T. hamatum*, *T. viride*, *T. pseudokoningii*, *T. polysporum*, and *T. virens* [33,34], which are the active ingredient of a vast amount of *Trichoderma*-based biofungicides [34]. These species of fungi have a significant impact on the development of plant diseases caused by *Rhizoctonia solani*, *Sclerotium rolfsii*, *Pythium aphanidermatum*, *Gaeumannomyces graminis* var. *tritici*, *Verticillium dahliae*, *Fusarium culmorum*, and *F. oxysporum* [34].

On the other hand, the potential of biological control and plant growth promotion of many members of rhizosphere-associated bacteria have also been explored for many decades, of which species belonging to the *Pseudomonas* and *Bacillus* genera have attracted the most attention. Particularly, the soil-ubiquitous *Bacillus* genus constitutes one of the main groups of beneficial microorganisms used against diseases caused by soil-borne phytopathogens [35]. Except for the pathogenic species *B. cereus* and *B. anthracis*, the genus *Bacillus* includes species with properties generally recognized as safe or with a classification of supposedly safe (GRAS/QPS) [36]. The species *B. subtilis,* and those from the operational group *B. amyloliquefaciens* (consisting of the soil-borne *B. amyloliquefaciens*, and plant-associated *B. siamensis* and *B. velezensis*) [37], are among the most used rhizobacteria in agriculture and have been extensively studied. These closely related species share properties, such as the production of cyclic lipopeptides with antimicrobial activity and the ability to colonize the rhizosphere and protect plants against phytopathogens. For instance, it has been determined that surfactin production is a determinant of root colonization [38,39], iturins and fengycins are known for their direct effect on plant pathogens [40,41,42], whereas systemic resistance can be induced by surfactins [43,44] and fengycins [45]. The importance of the relationship between beneficial strains of *Bacillus* spp. with plants and pathogens has merited the recent publication of several reviews viz., [28,46,47,48,49,50,51], in which it is possible to notice that *F. oxysporum* in one of the most commonly targeted pathogens.

With the arrival of Foc TR4 in Colombia, the need arose to investigate methods to counteract its dispersion, reduce its multiplication in the soil, and control the disease. Since there are no registered biopesticides for controlling FWB in Colombia, and in consonance with the emergency treatment that has been given to this phytosanitary problem, priority was given to determining the potential of biopesticides in the local market as alternatives to control FWB. Different publications have reported promising results by strains of different species from *Trichoderma* and *Bacillus* genera against both Foc R1 [52,53] and Foc TR4 [54,55], but, as far as we know, no studies have been carried out on biocontrol of Foc R1 and Foc TR4 with *Trichoderma* and *Bacillus*-based bioproducts in Colombia.

Currently, 395 bioproducts are registered with the Instituto Colombiano Agropecuario—ICA, of which 66 are manufactured based on microbial agents for the control of phytopathogens, which are mostly based on different *Trichoderma* (59%) and *Bacillus* species (23%). Other antagonists used in the bioproducts registered in the country are mainly *Paecilomyces lilacinus* and *Streptomyces racemochromogenes*. However, other *Trichoderma* and *Bacillus*-based bioproducts are registered as microbial inoculants with biofertilizer effects [56]. *F. oxysporum* is the biological target of 17% of these 66 bioproducts, but none are authorized to control Foc in bananas and not all registered bioproducts are available on the market.

With this context, the aim of this study was to determine the potential of available biofungicides in the Colombian market to control Foc R1. We tested the hypothesis of whether the *Trichoderma*-based bioproducts are more efficient than *Bacillus*-based ones against Foc, and we evaluated the effects of the biocontrol treatments on the physiological parameters of the plants growing in the presence of Foc R1. This work combined in vitro and in vivo evaluations of the antagonistic and biocontrol activity of bioproducts against Foc R1. The results of this study allow us to generate recommendations for further in vivo validations of potential biocontrol tools against Foc TR4.

## 2. Materials and Methods

### 2.1. Selection of Biofungicides

Three available *Trichoderma* and two *Bacillus*-based biofungicides registered with ICA to control *F. oxysporum* in various crops, and four more with known efficacy against Fusarium diseases through previous experiments with carnation, cape gooseberry, and tomato (Unpublished data), were included in this study (Table 1). The dosages of the bioproducts were chosen by considering the application of the same concentration of the active ingredient; thus, 1 × 10^6^ conidia mL^−1^ for *Trichoderma*-based bioproducts, and 1 × 10^7^ cells mL^−1^ for *Bacillus*-based bioproducts. The scheme for applying the bioproducts under in vivo experiments was designed according to their possible modes of action. Thus, the bioproducts were applied to the seedlings in the seedbed to facilitate the possible induction of resistance in the host, to stimulate the rhizospheric competence, and to promote endophytic colonization. The application of treatments to the soil before the transplant could favor the direct competition with the pathogen, the soil colonization, and facilitate direct modes of action against Foc.

### 2.2. Plant Material and Foc R1 Strain

Two-month-old banana seedlings of the susceptible Gros Michel type were used to evaluate the efficacy of bioproducts against the Fusarium wilt disease development in a pot experimental model under greenhouse conditions. Seedlings from Foc R1-resistant Cavendish type cv. Williams were included as a positive control. Plant propagation and supply were performed by the Eastern Catholic University (UCO), Rionegro, Antioquia, Colombia.

*F. oxysporum* f. sp. *cubense* race 1 IB strain (Foc R1 IB), supplied by the national banana research center (CENIBANANO), was used in the in vitro and in vivo experiments. The IB strain was isolated from Foc R1-susceptible Gros Michel banana plants showing typical symptoms of Fusarium wilt from plantations in Urabá, Colombia. PDA-plugs with mycelium and conidia of the IB strain were preserved in a glycerol sterile solution (20%) at −70 °C. The strain was activated in a potato dextrose agar culture medium (PDA—Oxoid^®^, Hampshire, UK) at 27 °C, and the second subculture was used as inoculum for the in vitro experiments, and as pre-inoculum to multiply the pathogen for the in vivo experiments.

### 2.3. Production of Foc R1 Inoculum for In Vivo Experiment

Mass production of Foc R1 IB was made following a solid-state fermentation model with wheat bran (72 g) in autoclavable plastic bags and moistened with 52 mL of sterile distilled water (SDW). The substrate in the bags was sterilized in an autoclave (two continuous cycles at 121 °C and 15 PSI). A suspension of microconidia from the IB strain was prepared by scraping a ten-day-old subculture in PDA, and the harvested biomass was added to sterile Potato Dextrose Broth (PDB—Difco^®^, Leeuwarden, The Netherlands) in 50 mL sterile Falcon^®^ (Mexico City, Mexico) tubes. The suspension was vortexed at 3000 rpm for 2 min, then it was filtrated by muslin cloth (0.5 mm mesh) and the microconidial suspension was adjusted to 1 × 10^7^ microconidia·mL^−1^. This suspension was injected into the sterile solid substrate (10 mL per bag) with the aid of a sterile syringe, then the substrate was massaged to homogenize the inoculum. The inoculated bags were incubated for 20 days at 27 °C and massaged every 5 days to promote uniform colonization. To obtain the inoculum of Foc R1 IB for the in vivo experiment, the colonized substrate from a bag was poured into 500 mL of SDW and was blended at low speed in a home-blender (Oster^®^, Wickliffe, OH, USA), then filtered through sterile muslin cloth to harvest the microconidial suspension free of mycelium. A total of 200 mL of conidial suspension (1 × 10^6^ microconidia·mL^−1^) was drenched into 2000 g of substrate (soil and rice husk mixture, 3:1) contained in a pot (4 Mil poly bags 35 cm height × 25 cm width) at 14 days before transplant (dbt) in order to obtain a final concentration of 1 × 10^5^ microconidia·g^−1^ of soil.

### 2.4. Treatments Application Scheme

The treatments (T1 to T9 described in Table 1) were applied in four moments: first, to the seedlings in the seedbed five days before transplant (5 mL seedling^−1^), second, to the soil contained in the pots five days before transplant (100 mL pot^−1^), third, to the soil again, immediately after transplanting (100 mL plant^−1^), and fourth, to the soil at one week after transplant (100 mL plant^−1^). While the treatments of *Trichoderma*-based biofungicides were applied adjusting the concentration shown in each label to a final concentration of 1 × 10^6^ conidia mL^−1^, the dosage of the *Bacillus*-based biofungicides was adjusted to 1 × 10^7^ cells mL^−1^. Plants growing in soil artificially inoculated with Foc R1 IB but not treated against the disease were included in the experiment as negative controls (T10), and plants growing in soil free of both Foc R1 IB and biological treatments were included as absolute controls (T12).

Although the doses of the biofungicides were calculated according to the label, quality control was carried out to estimate the concentration of viable active ingredients in terms of colony-forming units (CFU mL^−1^ or CFU g^−1^). To this end, 1 g or 1 mL, according to the formulation of the bioproducts, was diluted in 9 mL of Tween 80 sterile solution (0.1% *v*/*v*), it was vortexed at maximum speed for 2 min, and serial dilutions (100 µL in 900 µL of Tween 80 sterile solution at 0.1% *v*/*v*) were made and plated on solid culture media. Dilutions, from which colony counts of from 20 to 200 colonies for fungi and from 30 to 300 for bacteria were obtained, were taken into account for the estimations. While the suspensions from *Trichoderma*-based bioproducts were plated on PDA supplemented with Triton (0.1% *v*/*v*) and chloramphenicol (100 mg L^−1^), the suspensions from *Bacillus*-based bioproducts were plated on nutrient agar medium. Inoculated media were incubated for 48 h at 28 °C for bacteria counts, and seven days at 25 °C for fungi counts.

### 2.5. Experimental Design and Measured Disease Variables

The experiment was carried out under an unheated greenhouse with a temperature of 21.4 °C on average, 40.9 °C maximum, and 12.5 °C minimum, relative humidity of 60% on average, and photoperiod of 12 h light: 12 h darkness. The treatments were arranged under a randomized complete block experimental design. The experimental unit consisted of five plants and there were three repetitions per treatment. The entire experiment was replicated two times more over time, from 24 November 2021 to 9 February 2022 (first replicate), from 15 December 2021 to 25 February 2022 (second replicate), and from 11 February to 27 April 2022 (third replicate). The disease incidence, which expresses the proportion of plants showing typical symptoms of Fusarium wilt, and the disease severity, which expresses the intensity of the wilt symptoms, were recorded periodically. The scale from Dita et al. [57] with modifications was used for recording the disease severity (Appendix A). The area under the disease incidence and severity progress curves (AUDPC) was calculated using the trapezoidal rule.

With the aim to identify the inner symptoms of the disease and to measure its severity in the rhizome, all the alive plants were gently uprooted at the end of the experiment, and a longitudinal cut was made from the pseudostem to the rhizome on each plant, as was described by Dita et al. [57].

### 2.6. Physiological and Plant Growth Variables

To test the hypothesis that the application of biofungicides mitigates the negative effects caused by Foc R1 on the physiological processes of banana plants, the photosynthesis and gas exchange-related variables were measured in the plants of the biocontrol experiment under greenhouse as follows:

Chlorophyll *a* fluorescence. The maximum quantum efficiency of photosystem II-PSII (Fv/Fm), the PSII photochemical yield (Y(II)), and the electron transfer rate (ETR) were measured every 20 days through a modulated fluorometer miniPAM II (Heinz Walz^®^, Effeltrich, Germany). The measures were always taken from the third youngest expanded leaf on the central region of the leaf limb [58]. The readings were taken early in the morning, after a 20 min dark adaptation period.

Net assimilation rate (A). The variables associated with gas exchange [photosynthesis A (μmol m^−^² s^−^¹) and transpiration rate E (mol m^−^² s^−^¹)] were measured every 20 days using LI-COR^®^ LI-6800 portable photosynthesis system equipment (USA). To determine the optimal intensity of photosynthetically active radiation (PAR) for photosynthesis measuring in the banana plants, a light curve was made previously, applying increasing light intensities and measuring the photosynthesis response. The measurement conditions were as follows: 900 μmol·m^2^ s^−1^ of CO_2_ flux; 1200 mol·m^2^ s^−1^ of light and 60% of relative humidity.

Fluorescence and gas exchange variables were measured on three plants per treatment, sampling the representative plant per block (*n* = 3) at the beginning of the measurement, and always measuring the same plants.

Chlorophyll content was determined weekly based on the SPAD index using Minolta SPAD-502 leaf chlorophyll meter (Konica Minolta^®^, Osaka, Japan) equipment.

Growth parameters. The plant height from the pseudostem base to the insertion of the first expanded leaf, and the stem diameter measured with an analogous calibrator (Mitutoyo^®^, Kawasaki, Japan) at 5 cm height from the stem base, were recorded weekly. The measurements of chlorophyll index and growth variables were carried out weekly on three representative plants per block (nine plants per treatment, *n* = 9).

### 2.7. Molecular Detection of Foc in Plant Tissues

Fragments of rhizome tissue from both symptomatic and non-symptomatic plants were taken to confirm the presence of the pathogen in the inner tissue through a standardized procedure for the molecular detection of Foc R1 [59]. Fragments of tissue (nearly 5 mm^2^) were stored in sterile polypropylene tubes at −20 °C, lyophilized for 48 h, and then macerated in liquid nitrogen and the DNA extraction was performed using the protocol proposed by Daire et al. [60]. The DNA quality was verified by agarose gel electrophoresis and a quantitative estimation of DNA was made using the NanoDrop spectrophotometer (NanoDrop 2000, Thermo Scientific, Waltham, MA, USA). To determine the quality of the samples for PCR, the amplification of a gene that amplifies specific repetitive elements of musaceae (BrepF: 5′ GATTTTGTAGATTTTGGACACCG 3′ BrepR: 5′ GAATAACAAATATGCTCCAATACCC 3′) was performed following the methods described by Mansoor et al. [61]. The DNA extracted from Foc R1 pure culture and banana plant tissue inoculated with Foc R1 was used as a positive control, and DNA extracted from Kikuyu grass was considered a negative control.

The molecular marker SIX6b_210 (SIX6b_210_F: 5′ ACGCTTCCCAATACCGTCTGT 3′, SIX6b_210_R: 5′ AAGTTGGTGAGTATCAATGC 3′) was amplified by conventional PCR [59] on DNA samples extracted from plant tissue for the detection of Foc R1 in the rhizome samples. DNA extracted from the pure culture of Foc R1 and DNA extracted from the rhizome tissue of banana plants inoculated with Foc R1 were used as positive controls, and DNA from the pure culture of Foc R4T was included as a negative control. Amplification was performed under the following conditions: initial denaturation at 95 °C for 3 min, followed by 32 denaturation cycles at 95 °C for 15 s, banding at 56 °C for 30 s, an extension phase at 72 °C for 60 s, and a final extension cycle of 5 min at 72 °C. The PCR products were verified by agarose gel electrophoresis (1.4%), and the band size was determined with 1 Kb Plus DNA Ladder^®^ (Thermo Fisher Scientific, Waltham, MA, USA).

### 2.8. In Vitro Antagonism Test of Biopesticides against Foc Race 1

Dual confrontation tests were carried out on PDA medium (90 mm Petri dishes) between the antagonists, i.e., *Trichoderma* and *Bacillus*-based biofungicides and Foc R1. A suspension of each biopesticide was prepared in Tween 80 sterile solution (0.1% *v*/*v*) using the concentrations mentioned above for the greenhouse experiment and was vortexed at maximum speed for 2 min. A total of 10 µL suspension of Foc R1 IB (1 × 10^5^ microconidia·mL^−1^) harvested from a 7-day old solid culture, was dropped in the center of the PDA surface. Once the Foc drop dried, 10 µL suspensions of biopesticides were inoculated at two opposite ends 1 cm from the edge of the Petri dish, perpendicular to the Foc R1 inoculum. After the droplets dried, the dishes were incubated for 7 days under darkness and 25 °C.

Five Petri dishes were inoculated per treatment, and the dishes inoculated only with Foc R1 were the control group. A completely randomized design was used to arrange the plates in the incubator. The diameter of the Foc R1 colony was measured after a week of incubation, and the inhibition of Foc growth was calculated in comparison with the control.

### 2.9. Population of Microorganisms in the Soil

Rhizospheric-soil samples were taken one week after each application of treatments. to determine the viable population of the Foc R1 IB and the *Trichoderma* and *Bacillus* applied as bioproducts. A composite soil sample was collected from all pots for each treatment in 50 mL Falcon^®^ tubes and was stored at 4 °C until the time of sample processing. The sample was homogenized, and 5 g subsamples were placed in tubes with 45 mL of Tween 80 sterile solution (0.1% *v*/*v*), these were vortexed at maximum speed for 2 min, then serial dilutions in sterile Tween 80 were plated on Komada-agar medium for counting *Fusarium* colonies and on PDA supplemented with Triton X-100 (0.1% *v*/*v*) and chloramphenicol (100 mg L^−1^) for counting total fungi and *Trichoderma* spp. colonies. Lastly, to count similar colonies of *Bacillus* spp. on nutrient agar medium, the soil suspension sample was thermally shocked in a water bath at 90 °C for 15 min before being plated.

### 2.10. Data Analysis

Incidence data were submitted to a t-test for equality of variances between pairs of experiments, and, after the verification, the data from the three replicates of the experiment were pooled. Then, normality (Shapiro–Wilk, α = 0.05) and homoscedasticity tests (Bartlett, α = 0.05) were performed to determine the final incidence, AUDPC, and efficacy in reduction of AUDPC data before being submitted to ANOVA; here, each replicate of the experiment was considered a blocking factor. The media of the treatments were compared through Tukey’s test (α = 0.05). The exponential, logistic, and linear growth models were tested to describe the progress of the FW incidence curves using the PROC NLIN procedure of SAS Enterprise Guide v. 8.3, and the growth rate was calculated at the exponential phase of the incidence progress using the PROC REG procedure. The AUDPC of the *DII* from each replicate of the experiment was also submitted to the ANOVA and Tukey’s test for mean comparisons.

Data from the physiological and plant growth variables were also submitted to normality and homoscedasticity tests before the ANOVA, as described above. Then, a comparison of the media between treatments was carried out using Fisher’s least significant difference (LSD). In some cases of non-normality, the Kruskal–Wallis no parametrical test (α = 0.05) was performed using the Statgraphics Centurion XV software.

Data from the antagonism test between bioproducts and Foc R1 were square root-transformed to accomplish the assumptions of normality and homoscedasticity. These transformed data were submitted to an ANOVA test and the media of growth between treatments were compared through a Tukey’s test (α = 0.05).

Log_10_ data of the microorganism population in the soil were tested for normality and homoscedasticity. ANOVA was performed, and media of the treatments were compared through the Dunnet’s test (α = 0.05) using the population of Foc R1 as the control.

## 3. Results

### 3.1. Biocontrol of FWB under Greenhouse

#### 3.1.1. Concentration of the Active Ingredients in Bioproducts

The quality control to verify the concentration of the viable conidia of *Trichoderma* and viable endospores of *Bacillus* in the respective bioproducts showed a higher concentration than indicated on the label of the *Bacillus*-based biofungicides designated as T5, T6, T7, and T8. On the other hand, while a lower concentration of viable conidia was found in the *Trichoderma*-based biofungicides T1 and T2, the concentration of viable conidia in T3 was slightly higher, and that of T4 was similar to what was reported on the label, as well as in T9, in which the concentration corresponded to the concentration reported by the supplier company (Appendix A).

#### 3.1.2. Biocontrol Activity of Biofungicides against FWB

As shown in Table 2 and Figure 1, a high variation of the FWB incidence was observed in most of the treatments among the replicates of the entire experiment. However, the verification of equality of variances among the replicates allowed the pooling of the incidence data from the three replications of the experiment. Since the time after transplant in which the disease was recorded was not the same in the three replications of the experiment, the analysis of the incidence was performed in terms of progress per week. The logistic growth model $I(t)=\frac{K}{1+e^{-r(t-t0)}}$, in which *I*(*t*) is the incidence of the disease at time t, *K* represents the maximum incidence that the population can reach, *r* is the growth rate of the incidence, and *t*_0_ represents the time of inflection, fitted well to describe the progress of the incidence for all the treatments (Table 2). Thus, the average incidence from the nine experimental units and the predicted data from the model showed that T2, T3, and T7, highlighted by a less pronounced incidence curve S-shape (Figure 1), had a lower incidence progress rate (17, 21, and 10%, respectively), calculated at the exponential phase of the curves, and a high reduction in the AUDPC_INC_ (50, 37, and 74%, respectively) compared with the negative control (Table 2).

With respect to the analysis of the disease intensity index (*DII*), which considers the incidence as well as the severity of the disease for the calculation, the ANOVA of the first replicate of the experiment showed significantly lower AUDPC *_DII_* in banana plants treated with T2 (*T. harzianum*), T4 (*T. harzianum*, *T. koningii*, and *T. viride* consortium), and T7 (*B. amyloliquefaciens*, *Agrobacterium radiobacter*, and *B. pumilus* consortium), with efficacies of 71%, 48%, and 91% respectively, which represent the percentage of reduction of the FW intensity. In the second replicate, the progress of the *DII* was significantly reduced, only by T7 treatment, by 81%. Additionally, treatments T3 (*T. koningiopsis* Th003), T5 (*B. subtilis*), and T7 significantly reduced the progress of the *DII* in the third replicate by 69%, 59%, and 60%, respectively (Table 3). Although the high variation in the efficacy to reduce the FW intensity index by most biofungicides was also observed among the replications of the experiment, only the T7 treatment showed consistency and effectiveness among the replications (Figure 2, Appendix A).

#### 3.1.3. Molecular Detection of the Pathogen in Plant Tissue

A total of 316 plant samples from treatments with biofungicides (T1 to T9), the negative control (T10), and the absolute controls (T11, T12, and T13) from the first (172 samples) and second (144 samples) replicates of the entire experiment were analyzed. The extracted DNA from plant material showed good quality for amplification by conventional PCR, furthermore, when the Musaceae constituent gene was amplified, it was observed that 99% of the samples showed a band of 415 bp, corresponding to the expected size with the BrepI marker (Appendix A).

A total of 71.7% of samples from Foc R1-inoculated plants (T1 to T10) in replicate 1 showed a band of approximately 210 bp, corresponding to the amplification of the SIX6b marker (specific for the detection of Foc R1). The remaining 28.3% of samples did not amplify the SIX6b gene, despite having been inoculated with Foc R1, so, the pathogen was not detected in these plant material samples. It should be noted that only 20% of samples from T7 amplified the SIX6b gene (Figure 3).

On the other hand, 75.3% of samples from replicate 2 amplified the specific marker for Foc R1 SIX6b. In this case, as in replicate 1, the treatment T7 presented the highest number of negative samples (46.6%) for the SIX6b marker (Appendix A). In the case of plant 4 in treatment 6, two samples were evaluated, one corresponding to a symptomatic mother plant and another to an asymptomatic daughter plant, finding that the latter was negative for the specific marker for Foc R1.

At the end of the first replicate of the experiment, no healthy plants were found in treatments T4, T8, and T10 (negative control) (Appendix A). On the other hand, 80% of plants in the T7 treatment were apparently healthy, and the molecular analysis carried out on these samples did not detect the pathogen. In T1, T3, and T6, only 6.7% of plants did not show symptoms, and the pathogen was detected in all samples only from T6. The pathogen was also not detected in the healthy plants from T3 and T5. Half of the apparently healthy plants in T9 tested positive for Foc R1 detection by amplifying the marker for the SIX6b gene, so, these plants were asymptomatic even when the pathogen had colonized the inner tissue of the rhizome (Appendix A).

#### 3.1.4. Effect of Treatments on Physiological and Plant Growth Variables

##### Plant Physiology Response

Although four measurements of the variables were recorded during the trial period (around 70 days), one every 20 days, the description of the results obtained at 49 days after transplant are shown here, since the statistical differences were wider, preserving the trend observed over time.

The results of photosynthesis-related variables varied among the replications of the experiment, mainly in plants under the effect of *Trichoderma*-based biofungicides (Table 4). For instance, plants treated with the *Trichoderma*-based biofungicides T2 and T9 showed the highest values of CO_2_-Net Assimilation Rate (13.94 and 15.63 μmol m^−2^ s^−1^, respectively) in the first replicate of the experiment, but only T9 was statistically different to T10 [(negative control) 5.72 μmol m^−2^ s^−1^]. In the second replicate, the response trend of T2 was preserved, reaching the highest values of photosynthesis (8.30 μmol m^−2^ s^−1^, on average), which was significantly different from the negative control (2.01 μmol m^−2^ s^−1^). In contrast, in the third replication of the experiment, plants under T2 and T9 treatments showed values of 1.00 and 1.51 μmol m^−2^ s^−1^, respectively which were significantly lower than the values recorded in the negative control T10 (10.69 μmol m^−2^ s^−1^) (Table 4).

On the other hand, the response of the photosynthesis-related variables in plants treated with the *Bacillus*-based biofungicides T5 to T8 was consistent among the three replicates of the experiment and allowed us to determine that T7 showed promising results in the maintenance of the functioning of the photosynthetic apparatus in banana plants growing in the presence of Foc R1. Thus, the net assimilation rate values in plants treated with T7 were close, 13.43, 10.47, and 10.45 μmol m^−2^ s^−1^ in the first, second, and third replicate, respectively, and no significant differences were found compared with the absolute control T12 (plants free of both Foc R1 inoculum and biocontrol treatments) (Table 4). Plants treated with the *Bacillus*-based biofungicide T5 also showed high values of net assimilation rate, 11.03 and 9.96 μmol m^−2^ s^−1^ in R1 and R3, respectively (Table 4).

The response of plant transpiration was reciprocal to photosynthesis; high variation among *Trichoderma*-based treatments was also observed (Table 4). Thus, while the T2 and T9 treatments showed the highest transpiration values (0.005 and 0.007 mol m^−2^ s^−1^, respectively) in the first replicate of the experiment, low transpiration values were shown in the third replicate (0.0005 mol m^−2^ s^−1^ for both treatments), reciprocally with the decreasing of its photosynthetic activity. With respect to the second replicate, the transpiration values were like in the third replicate, reporting low transpiration in plants treated with *Trichoderma,* especially in T9. The highest transpiration rate was observed in plants free of both Foc R1 and biofungicides (T12) and was reciprocally associated with the highest values of photosynthetic rate (Table 4).

Plants under treatment T7 showed transpiration values between 0.0003 and 0.006 mol m^−2^ s^−1^ with no significant differences compared with the absolute control (T12), likely due to a lower rate of stomatal closure in these plants and, likewise, better gas exchange and easier transportation of water throughout the xylem.

Similar to the results of photosynthesis, plants treated with *Bacillus*-based T5 treatment showed high consistency among the replicates and reached transpiration values of 0.005, 0.001, and 0.003 mol m^−2^ s^−1^ in R1, R2 and R3, respectively, close to those values recorded in plants under the T7 treatment, and no significant differences were detected compared with the absolute control (T12), with values of 0.008, 0.00002, and 0.0051 mol m^−2^ s^−1^ in replicates 1, 2, and 3, respectively (Table 4**).**

At 49 days, the photochemical yield response of the PSII (Y(II)) was not significantly different among treatments in general, with the exception of T5, which showed a response that was significantly higher than the response in the negative control (T10), and T3, which showed the lowest value in the second replicate of the experiment (Table 4). The maximum quantum efficiency of photosystem II was also similar among the treatments in all three replicates of the experiment, with the exception of the response in T3 and T6 in the second and third replicates, respectively, with the significantly lowest values (Table 4).

Although no significant differences were found between most of the biocontrol treatments and the negative control (T10) for the electron transfer rate (ETR), the T4, T5, and T7 treatments showed the highest values (Appendix A).

The chlorophyll indices were stable until 42 days, after which values began to fall in treatments T8 and T10 (first replicate); T1, T9, and T10 (second replicate); and T1, T6, and T9, (third replicate) (Figure 4). Meanwhile, T7, T2, and T9 treatments (first replicate); T7 and T2 (second replicate), and T5 and T3 (third replicate) showed stable values of chlorophyll indices until the end of the experiment, which demonstrates the buffering capacity of chlorophyll degradation associated with affectations caused by the pathogen on the banana plant.

Figure 4 supports what was described above in terms of SPAD index or chlorophyll reduction at 49 days. For instance, the lowest affectations in chlorophyll content were shown by the T2, T7, and T9 treatments, with 20.0, 12.4, and 26.1%, respectively. In contrast, the chlorophyll content was reduced by 75.4% in plants from the negative control (T10) of the first replicate of the experiment, and considerably in the second and the third replicates. The T7 treatment showed the lowest rate of chlorophyll degradation (16.5%) in the second replicate of the experiment. The T5 treatment mitigated chlorophyll damage in the third replication, with a negative value of SPAD unit reduction (−7.2%).

Spearman ordinal correlation analysis was performed to determine whether the behavior of the physiological variables is explained to a significant extent by the disease progression. Although the correlation adjustments were not highly significant, it was noteworthy that increasing disease in plants tends to correlate negatively with values of the chlorophyll fluorescence-related variables (ETR, Fv/Fm, and Y(II)) (Appendix A).

##### Plant Growth Response

It was highlighted that plant height in the T2, T5, and T7 treatments was consistently less affected along the three replicates of the experiment as compared with the other treatments and with the negative control at 70 days (Figure 5). The plants from the negative control showed a decrease in the pseudostem diameter at 49 days, evidencing the dehydration caused by the disease (Appendix A). The lowest percentages of diameter reduction were observed in plants under the T2 and T7 treatments, as compared with the negative control (Figure 6), as is consistent with the other variables measured in this study. Thus, this suggests a protective effect by the microorganisms in biofungicides T2 and T7 on the physiological mechanisms of the banana plants facing the presence of Foc R1.

#### 3.1.5. In Vitro Antagonism against Foc Race 1

All the treatments significantly reduced the diametric growth of the Foc R1 colony. However, a higher inhibition was observed for the treatments containing *Trichoderma* as the active ingredient (T1–T4 and T9), with inhibitions from 78.2 to 85.1%, with the treatments T3, T4, and T9 being those with the highest effectiveness in the group of *Trichoderma*-based bioproducts. On the other hand, the highest amount of inhibition by the bacterial-based products was shown by T5 and T8. The lowest inhibition amount was observed with T6 at 48.5% (Figure 7, Appendix A).

## 4. Discussion

Biological control and the study of resistance in the host have been the most relevant strategies for the management of Foc in banana plants, with endophytic microorganisms in general, and strains from *Trichoderma*, *Pseudomonas*, and *Bacillus* genera in particular, as the most studied biocontrol agents [30].

The reduction in Foc growth by antagonists has been studied through in vitro dual confrontation tests and has been focused mainly on strains of different *Trichoderma* species. Evaluated strains of *T. viride*, *T. harzianum*, and *T. hamatum* inhibited Foc by from 80 to 84% [62]. A similar value of antagonism was found in the present work with the formulation based on *T. koningiopsis* Th003, which showed 85% inhibition, the other biofungicides reduced Foc growth from 78 to 82%. It was observed that strains of *Trichoderma* grew faster than Foc, suggesting an inhibitory effect through competition for space. On the other hand, an inhibition halo was observed in the interaction zone between Foc and *Bacillus* strains, suggesting the displaying of an antibiosis effect against Foc.

With respect to the effect of bioproduct application on plant physiology, the gas exchange variables allow us to analyze the real-time state of the plant related to the net assimilation of carbon (A) and transpiration (E), which can be affected by Foc R1 colonization in the roots and the subsequent vascular bundle occlusion since they limit the water and nutrients intake capacity stimulating a drought condition, so the plant responds early to the stomata closing [63]. In this work, this phenomenon was observed in the initial phases of the disease infection, without being reflected in damage to the chloroplasts, as was described by Dong et al. [64].

With the disease progression, the plants under treatment (T10 and T8) with the highest level of disease showed stomatal closure and, therefore, a lower rate of CO_2_ (*A*) assimilation. However, Fv/Fm values were not as variable between treatments as expected. In this sense, authors, such as Lawson et al. [65] and Tambussi et al. [66], demonstrated that the antenna-PSII complex integrity is not significantly affected early during water stress. The chlorophyll fluorescence-related variables are a good indicator of the plant stress responses since they measure the state of photosystem II, which, in the case of stress, dissipates excess light energy in different ways (fluorescence or heat) instead of harnessing it for photosynthesis [67]. The effective quantum yield of PSII (Y(II)) shows the probability of a photon being absorbed to drive the photochemical process. This process depends on the photosystem state and the leaf temperature [68].

In general, it is assumed that the normal Fv/Fm value is around 0.82, which indicates an optimal condition in the plant, and the values falling below 0.60 would indicate a loss of photosynthetic function [69]. At the end of the three replicates of the experiment, treatments that could maintain values above 0.70, despite infection damages, showed a stable trend in their response to Foc R1 infection, indicating that biofungicides may be associated with physiological condition maintenance and avoiding the oxidative damage in thylakoidal membranes, which results in photosynthetic efficiency decreasing, as mentioned by Cheng et al. [70].

The reductions in variables such as ETR indicate that the electrons captured during the photosynthesis light phase cease to be destined towards the dark phase, and, as a consequence, they must be dissipated in the form of heat, fluorescence, or by the photorespiratory response of severe stress in plants caused by pathogen presence [65]. In this work, T4, T5, and T7 maintained higher values of ETR indicating stability on thylakoidal electron flow conservation, unlike other treatments, where data distribution allowed us to observe low values, indicating a decrease in the ETR as an indicator of photosystem II thylakoid damage [71].

At this point, the non-stomatal limitations could be related to the production of reactive oxygen species (ROS) and the oxidative cascades that lead to chlorophyll degradation and chloroplast component damage [72,73], which is consistent with the results of the chlorophyll index variable (SPAD units) in the present study, which was significantly lower in plants with severe disease symptoms. The chlorophyll index (SPAD) measures the greenness of the leaves, allowing a correlation to be made between SPAD units and chlorophyll concentration. Accordingly, the damage caused by Foc can be related to chlorophyll degradation as a consequence of severe damage in the photosystems caused by mycotoxins, such as fusaric acid, or oxidative stress [74,75].

In nature, there are species of fungi and bacteria capable of colonizing plant tissues and inhibiting the affectation caused by plant pathogens; some can be higher than the pathogens, but others fail due to their low survival rate in the soil [76]. In this study, we evaluated nine treatments of bioproducts as biological control agents with active ingredients, five based on *Trichoderma* and four based on *Bacillus*.

As a consequence of physiological parameter alterations, plant growth can be severely affected, depending on damage degree and extent. The analysis of plant growth parameters, such as height and pseudostem diameter, showed differences among treatments, which were also associated with disease evolution and the inhibition caused according to the evaluated treatments. Reduction percentages lower than 10% and 20% for treatments T2 (*T. harzianum*) and T7 (consortium of *B. amyloliquefaciens*, *A. radiobacter*, and *B. pumilus*), respectively, were observed with respect to plant height, as well as a smaller reduction in pseudostem diameter in treatments T4 (*T. harzianum, T. koningii, T. viride*) and T7.

Different studies have shown that *Bacillus* species act as disease suppressors, with antifungal characteristics and plant growth promotion traits [58]. The strain of *Bacillus licheniformis* CSR-D4 significantly reduced the incidence of wilt disease caused by *F. oxysporum* f. sp. *cubense* Tropical Race 4 (Foc TR4) in banana plants evaluated under in vitro and in vivo conditions. Bioactive metabolites, such as iturin C, bacillomycin, and fengycin, produced by CSR-D4 inhibited the invasion of the pathogen, proving promising for use as an effective biocontrol agent in the management of Foc TR4 [12].

In the case of fungi, strains of *T. harzianum* have been studied to control *Fusarium* wilt in young banana plants treated with different *Glomus* species [77]. Growth parameters such as plant height and pseudostem diameter showed a significant increase when individual applications of *T. harzianum* and *Glomus* were made as compared with the control. The plant growth-promoting effect exerted by the species of *Trichoderma* is attributed to the production of some phytohormones, such as auxins, phosphorus solubilizing enzymes, and siderophores [78]. In plants inoculated with Foc R4T, the species of *Glomus* showed a positive effect related to seedling growth, and, in turn, healthy seedlings were obtained at the end of the trial. Combined treatments of *T. harzianum* and *Glomus* spp. resulted in a delay in the progression of the disease. However, a phenomenon of competition occurred when the application of the two biocontrol agents was carried out simultaneously.

Bubici et al. [30] reported efficacies in different studies on Foc bananas in field conditions with *Pseudomonas* (79%) and *Trichoderma* spp. (70%), and lower efficacies for *Bacillus* sp. (69%), mycorrhizae, and nonpathogenic strains of *Fusarium* (42–55%) finding that efficacies between pot experiments and field experiments showed similar efficacies on average (pot 65% and field 70%). However, Thangavelu and Gopi [53] reported isolates of *Trichoderma* sp. that decreased the disease and increased the growth parameters compared with non-treated plants inoculated with Foc R1, and the effectiveness was increased up to 100% when two isolates of *Trichoderma* sp. and *T. asperellum* were combined, and the growth parameters were increased up to 250%. In the present work, we found the highest effectiveness by the treatment of the consortium of two strains of *Bacillus* spp. and *A. radiobacter*, with efficacy values up to 74%, agreeing with the reports in other studies. However, in the case of *Trichoderma*-based bioproducts the highest efficacy was 44.3%, this is lower than the previous reports for *Trichoderma* spp. Considering these results, it is important to continue with studies to evaluate doses, volumes, and frequencies of application for promising biofungicides and the search for other microorganisms that could increase the effectiveness, as well as the construction, of consortia of microorganisms with different modes of action.

There has been evidence for more than 30 years that *A. radiobacter* is an effective biological control agent with the strain K84. Coincidentally, this is one of the components of the active ingredient of the most promissory treatment in this research. K84 is classified as non-pathogenic for its lack of capacity to form galls, and its mode of action against *Agrobacterium tumefaciens* relies on the production of one highly specific anti-agrobacterial antibiotic, Agrocin 84, and also ALS 84, an antibiotic type of compound related to siderophore production [79,80]. Although its mechanism of action against other types of phytopathogens is still unclear, it has been reported to control *Erwinia* sp. and *Pseudomonas* sp. by the action of antibiotic compounds [81]. Additionally, there are reports of old data that already described the inhibition of Foc growth by *A. radiobacter* under in vitro conditions [82].

Some of the microorganisms evaluated and reported as biological control agents with promising results for the control of wilting caused by Foc include genera of fungi as well as bacteria that have shown indirect modes of action, like induction of systemic resistance in plants, production of fungitoxic metabolites, stimulation from competition for essential nutrients [47], as well as direct modes of action, such as antibiosis, mycoparasitism, and production of cell wall degrading enzymes (CWDEs) [83]. Studying and identifying the mechanisms triggered by the interaction between the biological control agents and the plant pathogens could be a step toward creating the auspicious conditions for their success or improving the strategies of biological control [84].

In the present study, we found that the biofungicides that showed higher inhibition of the Foc vegetative growth were T3, T9, T4, and T8, and that a reduction in the viable population of the pathogen in the soil was exerted by T3, T1, T4, and T9. These conditions, where the direct mechanisms of action could be displayed against the pathogen, were not the best treatments for reducing the disease in the in planta experiment, where T7 and T2 were the most effective treatments for controlling Fusarium wilt. Additionally, T7 biofungicide prevented the inner colonization of plant tissues by the pathogen in the 80% of plants inoculated with Foc, suggesting that the most effective control of the disease could be associated with other mechanisms different than direct ones, such as induction of resistance in the host.

Akila et al. [52] found a correlation between the decrease in the incidence of the disease caused by Foc R1 and the induction of the defense-related enzymes peroxidase (PO) and polyphenol oxidase (PPO) when combining botanical extracts and BCA *P. fluorescens* and *B. subtilis.* Li et al. [55] reported that the application of an alkaline fertilizer and biocontrol fungi had a synergistic effect in the reduction of disease related to the peroxidase (PO), catalase, and superoxide dismutase (SOD) activity associated with the antioxidant system of the plant. Lin et al. [85] found that when the plant defense system is activated by BCA it can improve its ability to resist the colonization of the pathogen, similar to the observed effect of the T7 treatment in the present study. However, these hypotheses related to the modes of action of T7 members deserve further studies to elucidate biocontrol performance, since it was the most promising treatment against Foc R1 in this study. In addition, currently, we are validating the results obtained in the present study with the Fusarium wilt caused by Foc R4T under biosecurity conditions.

## 5. Conclusions

In this study, the treatments that showed high efficacy in the inhibition of diametral growth and the reduction of the viable population of the pathogen in the soil were not the best at reducing the disease under the in planta assays, suggesting that the control of the disease was related to indirect mechanisms of action, like induction of systemic resistance in the plant. Although available bioproducts in the market were identified with the potential to reduce the disease and to mitigate the biotic stress in the plant by the presence of Foc R1, the effect of different cost-effective doses and the frequency of applications of biocontrol agents deserve further research. Continuing with the search for more efficient microorganisms and the evaluation of microbial consortia with different modes of action that could improve the control of the disease, and characterizing the mechanisms of action by the most effective treatments, are also important issues to be studied, with the aim of understanding the interactions between the host, the pathogen, and the biocontrol agent governing the effectiveness against Foc infections.

## Figures and Tables

**Figure 1 jof-10-00419-f001:**
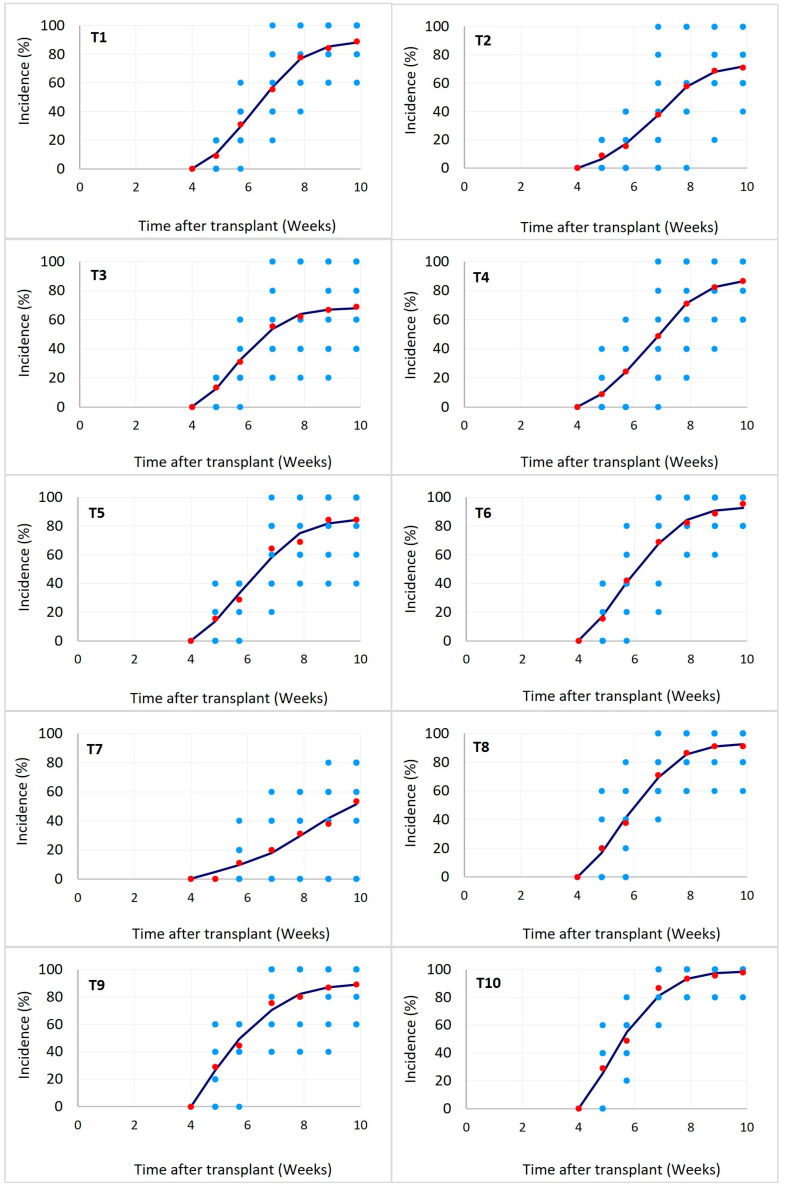
Effect of microbial biofungicides (T1–T9) on Fusarium wilt incidence progress in banana plants. Blue marks represent the original data. Red marks represent the average incidence (*n* = 9). The black line shows the predicted incidence by the logistic growth model as fitted for each treatment. The parameters of the model are shown in Table 2. T10: negative control.

**Figure 2 jof-10-00419-f002:**
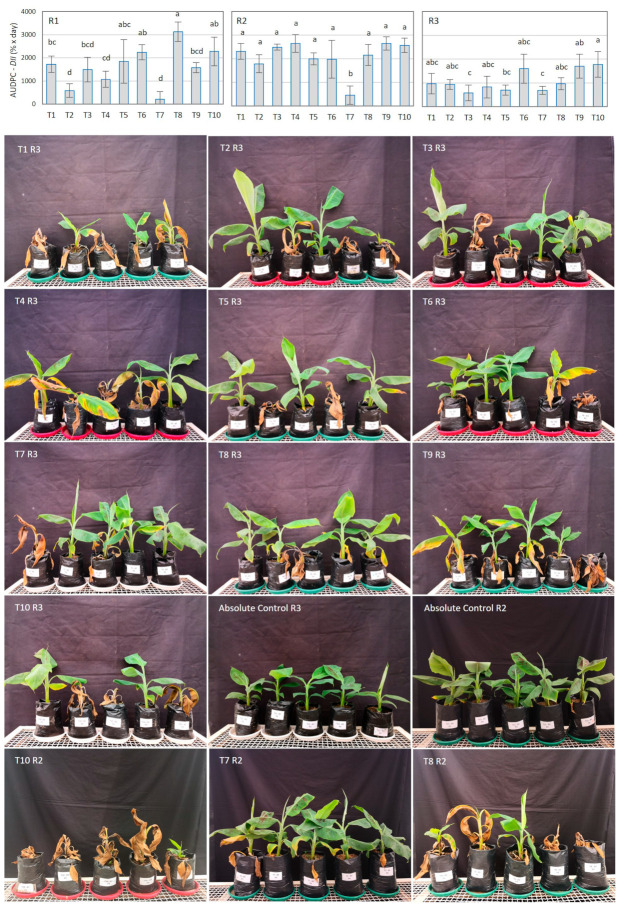
Effect of microbial biofungicides on Fusarium wilt severity in banana plants. The graphics in the top panel show the effects of biofungicides on the area under the disease intensity index progress curve (AUDPC DII) at the end of each replicate (R1–R3) of the experiment (77, 72, and 68 days after transplant for R1, R2, and R3, respectively). Bars on the columns represent the standard deviation of the mean (*n* = 3). Columns with the same letter are not significantly different according to Tukey’s test (α = 0.05). Biofungicides: T1–T9. Negative control: T10. The images show the representative state of the plants at the end of the experiment in replicates 2 (R2) and 3 (R3) of the entire experiment.

**Figure 3 jof-10-00419-f003:**
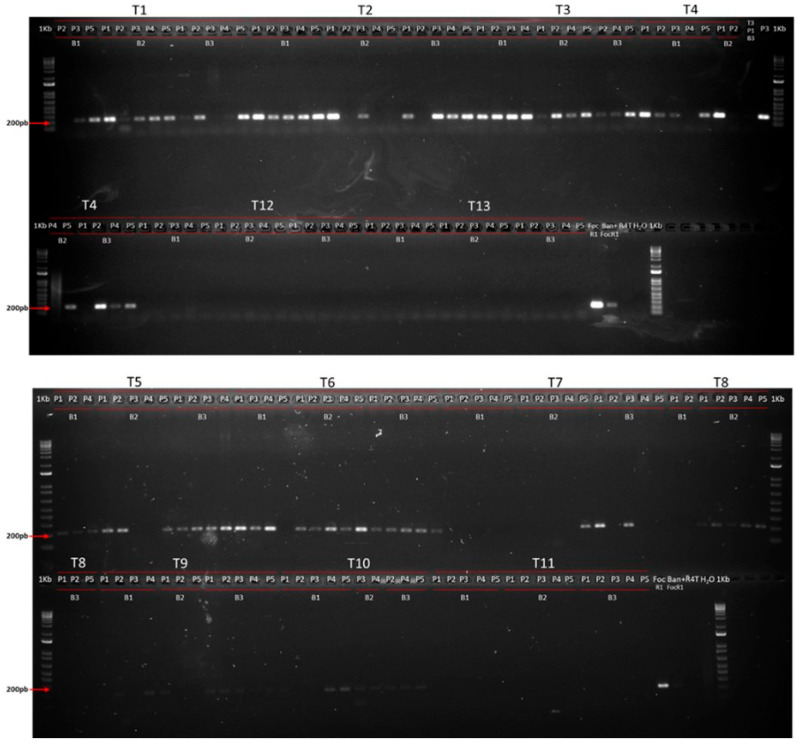
Electrophoresis in agarose gel (1.4%) used for verification of PCR amplification of the marker SIX6b (specific for Foc R1) on DNA extracted from banana plant samples in the first replicate of the experiment.

**Figure 4 jof-10-00419-f004:**
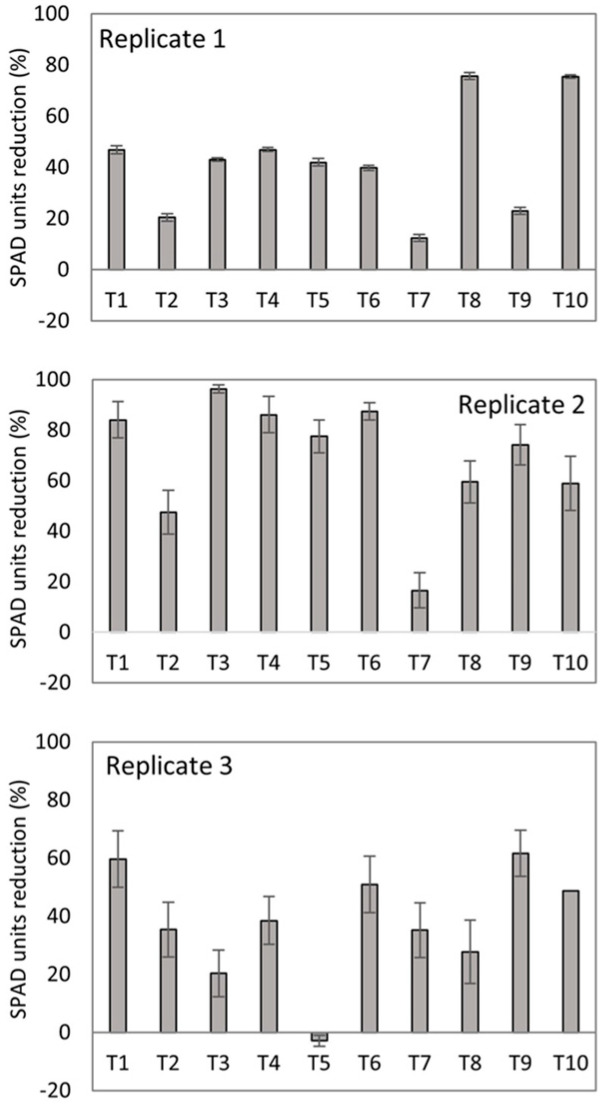
Chlorophyll index reduction in banana plants at 70 days after transplantation in soil artificially inoculated with Foc R1 and treated with biofungicides. Bars on the columns represent the standard deviation of the mean (*n* = 3).

**Figure 5 jof-10-00419-f005:**
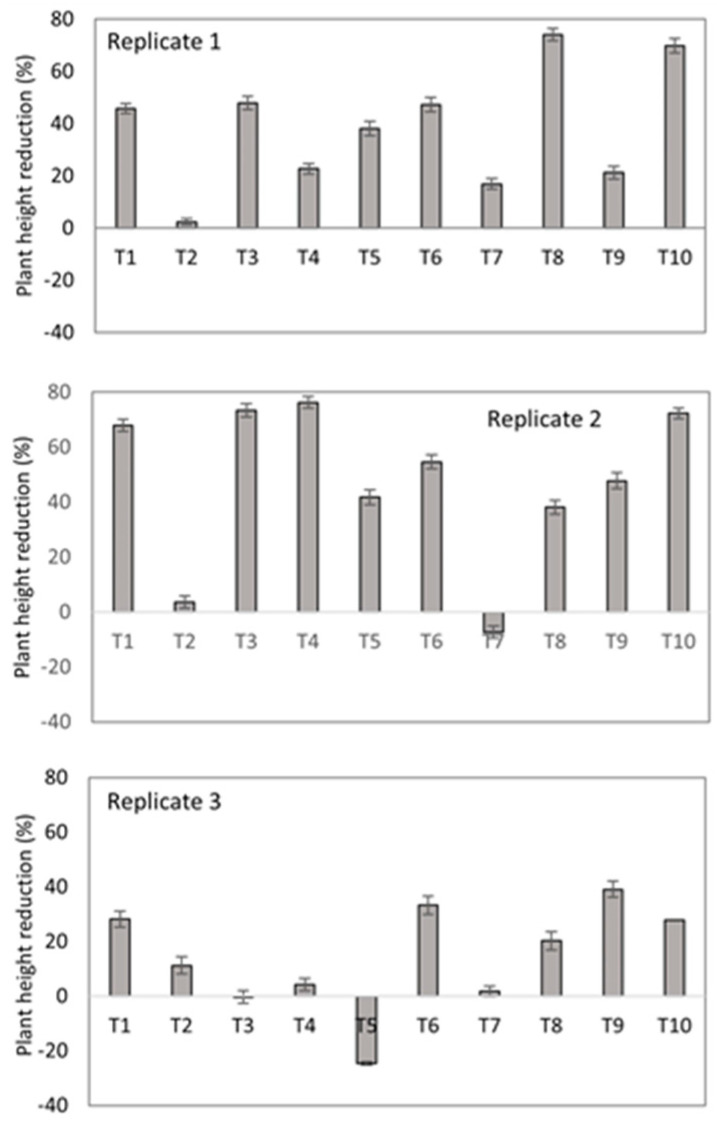
Plant height reduction of banana plants grown in soil artificially inoculated with Foc R1 at 70 days after transplant. Bars on the columns represent the standard deviation of the mean (*n* = 3).

**Figure 6 jof-10-00419-f006:**
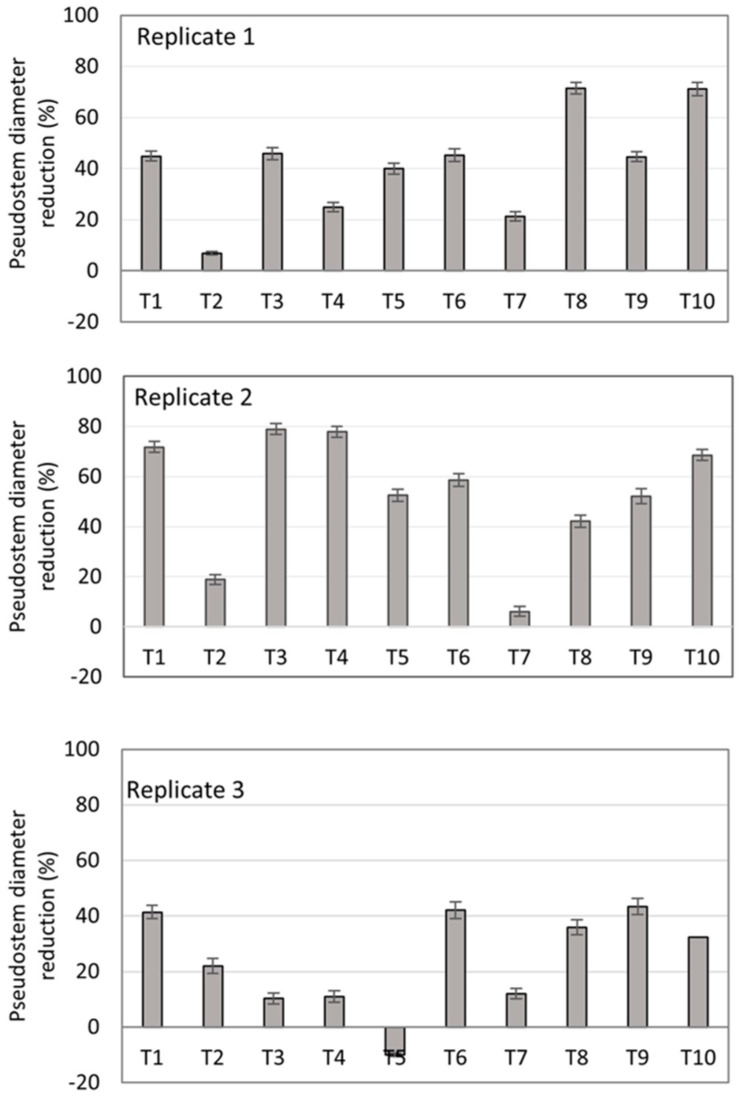
Pseudostem diameter reduction of banana plants at 70 days after transplantation in soil artificially inoculated with Foc R1. Bars on the columns represent the standard deviation of the mean (*n* = 3).

**Figure 7 jof-10-00419-f007:**
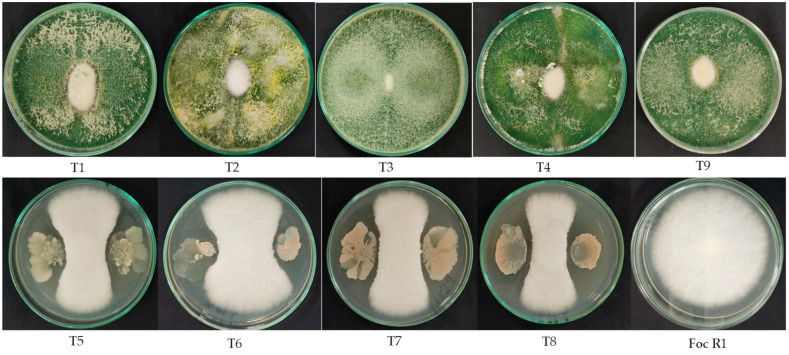
In vitro antagonism test of the microbial fungicides against Foc R1. The inhibition of Foc growth was calculated on the basis of the diameter of the colony measured one week after incubation at 25 °C.

**Table 1 jof-10-00419-t001:** Description of biofungicides selected to be evaluated against Foc R1.

Treatment	Active Ingredient	Target Pathosystem	Formulation	Concentration (CFU g^−1^ or mL^−1^)
T1	*T. harzianum*	*F. oxysporum*—Carnation	WP	1.0 × 10^8^
T2	*T. harzianum*	*F. oxysporum*—Carnation	WP	2.0 × 10^8^
T3	*T. koningiopsis*	*F. oxysporum*—Tomato	WDG	1.0 × 10^9^
T4	*T. harzianum, T. koningii, T. viride*	Various soil-borne phytopathogens	WP	1.0 × 10^8^
T5	*B. subtilis*	*F. oxysporum*—tomato	SC	5.0 × 10^9^
T6	*B. subtilis, B. pumilus, B. thuringensis* var. *kurstaki, B. amyloliquefaciens*	*F. oxysporum*—tomato, cape gooseberry, lulo, tree tomato, chili pepper, pepper, and egg plant	SC	4.0 × 10^8^
T7	*B. amyloliquefaciens, Agrobacterium radiobacter, B. pumilus*	Known performance against *F. oxysporum*—Carnation at experimental level	SC	2.5 × 10^8^
T8	*B. amyloliquefaciens*	*Mycosphaerella fijiensis*—banana	SC	1.0 × 10^9^
T9	Mixture of *Trichoderma harzianum* strains (no formulated)	Applied by commercial banana growers	Unformulated conidia	

WP: Wettable Powder; WDG: Water-Dispersible Granule; SC: Suspension Concentrate.

**Table 2 jof-10-00419-t002:** Effect of soil treatment with biofungicides on Fusarium wilt incidence in banana plants.

	Parameters ^a^			Statistics ^b^								
Treatment	*K*	r	t_0_	F	Pr > F	R^2^	Growth Rate (%) ^c^	Final INC (%) ^d^	CV ^e^ _INC_	AUDPC-INC ^f^ (% x Weeks)	CV ^g^ _AUDPC_	AUDPC Reduction (%) ^h^
T1	89.3	1.28	6.6	264.38	<0.0001	0.999	22.7	88.89	16.35	322.70	24.88	29.90 (4.3) bc
T2	73.7	1.2	7.0	90.13	<0.0001	1.000	17.3	71.11	28.51	243.65	51.32	49.72 (7.0) b
T3	68.2	1.4	6.1	95.88	<0.0001	1.000	21.1	68.89	38.71	292.06	41.31	37.12 (7.0) bc
T4	88.54	1.19	6.82	127.38	<0.0001	1.000	20.8	86.67	19.99	298.89	47.42	37.38 (7.6) bc
T5	85.11	1.23	6.36	190.47	<0.0001	0.997	24.8	84.44	25.88	337.94	35.61	28.22 (8.3) c
T6	93.60	1.23	6.20	300.01	<0.0001	0.999	22.2	95.56	9.23	379.37	27.95	20.77 (6.4) c
T7	64.48	0.77	8.22	37.53	<0.0001	0.991	10.1	53.33	62.19	128.10	77.33	73.90 (6.5) a
T8	93.32	1.30	6.17	301.30	<0.0001	1.000	23.1	91.11	15.95	394.44	21.52	19.47 (5.2) c
T9	89.69	1.09	5.81	216.27	<0.0001	0.999	23.5	88.89	19.84	419.05	34.05	14.57 (5.1) c
T10	98.74	1.30	5.83	556.26	<0.0001	0.999	29.1	97.78	6.82	462.06	20.37	-

^a^ Parameters describing the logistic growth model for FW incidence. K is the maximum incidence that the population can reach, r is the growth rate of the incidence per week, and t0 represents the week in which the curve reaches the inflection. ^b^ Statistics describing the fitting of the logistic growth model of the incidence. F value, significance of the model, and determination coefficient. ^c^ The growth rate of the incidence curve at the exponential phase calculated based on the lineal model Y = b × X + a. ^d^ The average incidence of banana plants showing typical symptoms of Fusarium wilt at 10 weeks after transplant (*n* = 9). ^e^ Coefficient of variation of the final incidence of Fusarium wilt. ^f^ The average of the area under the Fusarium wilt incidence progress curve calculated based on the percentage of the incidence per week (*n* = 9). ^g^ Coefficient of variation of the area under the FW incidence progress curve. ^h^ The average reduction in the progress of the FW incidence compared with the negative control (T10) calculated based on the AUDPC value. Value in parenthesis represents the standard error (*n* = 9). Treatments sharing the same letter are not significantly different according to Tukey’s test (α = 0.05).

**Table 3 jof-10-00419-t003:** Effect of soil treatment with biofungicides on Fusarium wilt incidence on banana plants.

Treatment	Reduction of AUDPC *_DII_* (%) per Replicate *		
	1st	2nd	3rd
T1	18.6 ± 36.2	8.1 ± 23.3	47.3 ± 10.7
T2	70.9 ± 18.8	28.8 ± 22.3	46.3 ± 5.8
T3	30.7 ± 34.2	2.1 ± 11.3	69.4 ± 12.6
T4	47.8 ± 31.4	−5.4 ± 28.5	50.7 ± 17.6
T5	6.5 ± 68.1	21.2 ± 17.0	59.2 ± 11.7
T6	−0.9 ± 14.1	19.6 ± 41.5	−5.2 ± 77.9
T7	90.9 ± 14.6	81.4 ± 14.9	60.0 ± 10.3
T8	−46.1 ± 54.5	15.9 ± 14.2	41.1 ± 25.4
T9	26.9 ± 25.0	−5.2 ± 24.5	1.8 ± 20.1

* 1st, 2nd, and 3rd replication of the entire experiment. Each value represents the average efficacy compared to the negative control ± the standard deviation (*n* = 3).

**Table 4 jof-10-00419-t004:** Effect of biofungicides on physiological responses in banana plants from the three replicates of the entire experiment (R1 to R3).

	Net Assimilation Rate (A)						Transpiration Rate (E)						PSII Photochemical Yield						Quantum Efficiency of PSII					
	(μmol m^−^² s^−^¹)						(mol m^−^² s^−^¹)						Y(II)						(Fv/Fm)					
	49 Days after Transplant																							
	R1		R2		R3		R1		R2		R3		R1		R2		R3		R1		R2		R3	
T1	9.74	^abc^	5.54	^abc^	4.95	^abcde^	0.004	^bcd^	0.000	^a^	0.002	^a^	0.83	^a^	0.13	^ab^	0.08	^a^	0.56	^a^	0.80	^a^	0.55	^a^
T2	13.94	^ab^	8.30	^ab^	1.00	^e^	0.005	^abc^	0.001	^a^	0.001	^a^	0.14	^a^	0.11	^ab^	0.06	^ab^	0.83	^a^	0.77	^a^	0.54	^a^
T3	8.39	^abc^	0.00	^c^	3.12	^bcde^	0.003	^bcd^	0.000	^a^	0.001	^a^	0.10	^a^	0.00	^c^	0.08	^a^	0.81	^a^	0.00	^b^	0.55	^a^
T4	10.42	^abc^	5.58	^abc^	2.75	^cde^	0.004	^bcd^	0.000	^a^	0.001	^a^	0.14	^a^	0.07	^bc^	0.13	^a^	0.84	^a^	0.53	^a^	0.83	^a^
T5	11.03	^abc^	4.08	^abc^	9.96	^abc^	0.005	^abcd^	0.001	^a^	0.003	^a^	0.13	^a^	0.18	^a^	0.13	^a^	0.83	^a^	0.79	^a^	0.82	^a^
T6	6.69	^bc^	4.67	^abc^	0.00	^e^	0.002	^cd^	0.000	^a^	0.000	^a^	0.13	^a^	0.15	^ab^	0.00	^b^	0.75	^a^	0.81	^a^	0.00	^b^
T7	13.43	^abc^	10.47	^a^	10.45	^abc^	0.006	^abc^	0.000	^a^	0.005	^a^	0.14	^a^	0.15	^ab^	0.12	^a^	0.83	^a^	0.81	^a^	0.80	^a^
T8	4.89	^c^	4.96	^abc^	8.71	^abcd^	0.001	^d^	0.000	^a^	0.004	^a^	0.08	^a^	0.12	^ab^	0.12	^a^	0.55	^a^	0.80	^a^	0.82	^a^
T9	15.63	^a^	5.51	^abc^	1.51	^de^	0.007	^ab^	0.000	^a^	0.001	^a^	0.15	^a^	0.11	^ab^	0.08	^a^	0.83	^a^	0.55	^a^	0.51	^a^
T10	5.72	^bc^	2.01	^bc^	10.69	^ab^	0.001	^d^	0.000	^a^	0.004	^a^	0.09	^a^	0.07	^bc^	0.12	^a^	0.78	^a^	0.52	^a^	0.82	^a^
T12	16.38	^a^	8.29	^ab^	11.05	^a^	0.008	^a^	0.000	^a^	0.005	^a^	0.14	^a^	0.15	^ab^	0.13	^a^	0.84	^a^	0.84	^a^	0.82	^a^
	70 days after transplant																							
	R1		R2		R3		R1		R2		R3		R1		R2		R3		R1		R2		R3	
T1	4.43	^a^	0.00	^b^	0.00	^c^	0.001	^a^	0.000	^b^	0.001	^a^	0.05	^ab^	0.00	^c^	0.03	^cd^	0.56	^ab^	0.00	^c^	0.25	^bc^
T2	3.78	^a^	3.59	^ab^	2.32	^ab^	0.000	^a^	0.001	^ab^	0.001	^a^	0.05	^ab^	0.04	^abc^	0.09	^abc^	0.50	^ab^	0.27	^bc^	0.54	^ab^
T3	5.37	^a^	0.00	^b^	0.33	^bc^	0.001	^a^	0.000	^b^	0.001	^a^	0.04	^ab^	0.00	^c^	0.07	^bcd^	0.53	^ab^	0.00	^c^	0.45	^abc^
T4	3.19	^a^	2.63	^ab^	0.83	^bc^	0.000	^a^	0.001	^b^	0.001	^a^	0.04	^ab^	0.07	^abc^	0.11	^abc^	0.48	^ab^	0.38	^abc^	0.77	^a^
T5	4.40	^a^	0.00	^b^	1.43	^bc^	0.001	^a^	0.000	^b^	0.001	^a^	0.04	^ab^	0.00	^c^	0.12	^ab^	0.51	^ab^	0.00	^c^	0.78	^a^
T6	0.15	^a^	0.32	^b^	0.00	^c^	0.000	^a^	0.000	^b^	0.000	^a^	0.01	^ab^	0.02	^bc^	0.00	^d^	0.27	^ab^	0.24	^bc^	0.00	^c^
T7	0.72	^a^	7.56	^a^	0.69	^bc^	0.000	^a^	0.003	^a^	0.002	^a^	0.07	^a^	0.11	^a^	0.08	^abcd^	0.57	^ab^	0.79	^a^	0.53	^ab^
T8	0.00	^a^	3.89	^ab^	0.61	^bc^	0.000	^a^	0.002	^ab^	0.001	^a^	0.00	^b^	0.08	^abc^	0.14	^ab^	0.00	^b^	0.35	^abc^	0.80	^a^
T9	6.07	^a^	3.05	^ab^	2.22	^abc^	0.001	^a^	0.001	^ab^	0.001	^a^	0.04	^ab^	0.06	^abc^	0.00	^d^	0.56	^ab^	0.51	^ab^	0.00	^c^
T10	0.00	^a^	0.00	^b^	0.50	^bc^	0.000	^a^	0.000	^b^	0.001	^a^	0.00	^b^	0.00	^c^	0.07	^bcd^	0.00	^b^	0.00	^c^	0.51	^abc^
T12	4.06	^a^	4.58	^ab^	3.95	^a^	0.000	^a^	0.001	^ab^	0.001	^a^	0.07	^a^	0.11	^ab^	0.16	^a^	0.85	^a^	0.81	^a^	0.80	^a^

Each value represents the average from three sampled plants (*n* = 3). Treatments sharing the same letter are not significantly different according to Fisher’s minimum significant difference (LSD, α = 0.05).

## Data Availability

The datasets generated during and/or analyzed during the current study are available from the corresponding author upon reasonable request.

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
