# Peer review of "Microbial-Based Biofungicides Mitigate the Damage Caused by Fusarium oxysporum f. sp. cubense Race 1 and Improve the Physiological Performance in Banana"

_jof, 2024, doi:10.3390/jof10060419_

Round 1

Reviewer 1 Report

A minor revision of the manuscript

Introduction

Line 93: please change “Fusarium culmorum, and Fusarium oxysporum” to “F. culmorum, and F. oxysporum

Materials and Methods

Line 169: what did “Foc R1 IB” mean?

Line 200: please provide more information about “WP”, “WG” and “SC” in Table 1. And “CFU g or ml-1”? Did you mean “CFU g-1 or ml-1”?

It is better to use uniform units in this manuscript, such as “2 min” in line 175 and “two minutes” in line 217.

Line 319: change “two minute” to “2 min” or “two minutes”, or something else.

Results

Line 354: “cfu g-1 or cfu ml-1”?

Line 428-476: A title was needed in Figure 2.

Line 483: a comma was needed between “experiment” and “no”.

Line 598: “en the second”?

Discussion

Line 643: please change “In vitro” to “in vitro”.

Reference

Please revise the journal names with uniform writing, such as line 862, line 891, lines 894-895, and so on.

Please check the abbreviation of “Fusarium oxysporum f. sp. cubense (Foc)” in the whole manuscript, some were italic, and others were not, for example line 14, line 17, line 39, line 40, etc.

Reviewer 2 Report

Dear colleagues, the work is of interest in terms of studying plant protection products. It would be nice if you could make suggestions about the possible mechanisms of action of these treatments. You have used bactenial and fungal treatments. Are there any differences in the mechanisms of their action?,

Lilie "according to the formulation of the bioproducts, was diluted in 9 ml of Tween 80 sterile solution (0.1 % v/v), it was vortexed at maximum speed for two minutes",   - why was Twin needed? What is the role?

Line 220-221 "While the suspensions from Trichoderma-based bioproducts were plated on PDA supplemented with Triton (0.1% v/v) and chloramphenicol (100 mg L-1)" - What fre the roles of Triton and chloramphenicol,
Line 785-767 , "the treatments that showed high efficacy in the inhibition of diametral  growth and the reduction of the viable population of the pathogen in the soil, were not 786 the best reducing the disease under in planta assays, suggesting that the control of the  disease was related to indirect mechanisms of action - Which ones exactly?

Reviewer 3 Report

There are many scientific studies indicating the possibility of using fungi of the genus Trichoderma and Bacillus for the biological control against various diseases of trees and crops. Certain methods of their practical use in crop breeding have already been developed. The aim of the research presented in the current manuscript was to test the effectiveness of microbial-based biofungicides against Fusarium wilt of banana (FWB). It is a very dangerous disease caused by Fusarium oxysporum f. sp. cubense (race 1). Biofungicides based on five Trichoderma species and four Bacillus species were evaluated in pot experiments. Tests were also carried out in vitro in dual confrontation test. The methodology was well presented, some experiments were based on the methodology used by other authors (these works are cited appropriately). Interesting results were obtained, although the evaluation and interpretation of some of them were not easy. Research results regarding the effect of bioproducts application on the plant physiology (e.g. assimilation, transpiration) are also of great value. The manuscript is carefully prepared. Only in some places the text is too imprecise. This applies, among others, to the title "PGPR and Trichoderma spp.-based biofungicides..."a/ the abbreviation PGPR is widely known, but it is difficult to find a reason for its use in this manuscript. The best evidence is that apart from the title it was not used in the text, b/the sentence structure is unclear, whether it should be ‘PGR- and Trichoderma spp. -based...?The title needs to be changed. The manuscript should be published in JoF after minor revisions (see Remarks). [Please note - there is an error in page numeration].

Line 2 – title requires change (PGPR and Trichoderma spp.-based biofungicides)

Line 33 it should be… 4].

Line 43 The sentence should not start with 'And ……'

Line 81 You use spp. very often - but it is not justified everywhere. If the context shows that these are different species of a given genus (as in line 81), it is better to use 'different Trichoderma species'. Please take this into account throughout your manuscript

Line 83 chitobioses – do they belong to the group of enzymes?

Line 91 it should be Rhizoctonia solani

Line 314 consider revising this sentence (treatments.)

Line 321 Fusarium – it should be in italic

Line 354 'viable spores of Bacillus' - does Bacillus produce spores like in fungi?, the text requires clarification of what is meant

Line 620 Trichoderma could limit the pathogen not on a biochemical but physical basis (rapid growth)

Line 640 ‘being the endophytic microorganisms’ – this requires specifying which microorganisms are meant (is Trichoderma classified as an endophyte?)

Line 647 The problem of what caused such high Foc inhibition by Trichoderma should be included in the discussion, and the rapid growth of Trichoderma in vitro should be emphasized as the primary cause of inhibition - unless the authors have a different view

Line 660 it should be Lawson et al. [66] instead of Lawson [66]

Line 710-711 consider revising this sentence (produced CSR-D4 demonstrated ?)

Round 2

Reviewer 2 Report

Dear colleagues
Thanks for the answers, the article has become more understandable.

Speaking of a not very successful design, I meant the lack of photos. With their inclusion, the results became clearer. It can be seen how the Trichoderma suppresses the development of the fungus.
Regarding the mechanisms of disease resistance, you are absolutely right, it really can be systemic induced resistance.
I wonder what mechanism Triton X-100 uses to limit the growth of Trichoderma colonies?